# Role of microRNAs in Venous Thromboembolism

**DOI:** 10.3390/ijms21072602

**Published:** 2020-04-09

**Authors:** Vânia M. Morelli, Sigrid K. Brækkan, John-Bjarne Hansen

**Affiliations:** 1K.G. Jebsen Thrombosis Research and Expertise Center (TREC), Department of Clinical Medicine, UiT—The Arctic University of Norway, N-9037 Tromsø, Norway; sigrid.brakkan@uit.no (S.K.B.); john-bjarne.hansen@uit.no (J.-B.H.); 2Division of Internal Medicine, University Hospital of North Norway, N-9037 Tromsø, Norway

**Keywords:** microRNAs, venous thromboembolism, venous thrombosis, biomarker, animal model

## Abstract

MicroRNAs (miRNAs) are non-coding RNAs that execute their function by targeted downregulation of gene expressions. There is growing evidence from epidemiological studies and animal models suggesting that the expression level of miRNAs is dysregulated in venous thromboembolism (VTE). In this review, we summarize the current knowledge on the role of miRNAs as biomarkers for VTE and provide general insight into research exploring the modulation of miRNA activity in animal models of venous thrombosis. Up to now, published studies have yielded inconsistent results on the role of miRNAs as biomarkers for VTE with most of the reports focused on diagnostic research. The limited statistical power of the individual studies, due to the small sample sizes, may substantially contribute to the poor reproducibility among studies. In animal models, over-expression or inhibition of some miRNAs appear to influence venous thrombus formation and resolution. However, there is an important gap in knowledge on the potential role of miRNAs as therapeutic targets in VTE. Future research involving large cohorts should be designed to clarify the clinical usefulness of miRNAs as biomarkers for VTE, and animal model studies should be pursued to unravel the role of miRNAs in the pathogenesis of VTE and their potential as therapeutic targets.

## 1. Introduction

Venous thromboembolism (VTE), a collective term for deep vein thrombosis (DVT) and pulmonary embolism (PE), is the third most common cardiovascular disease with an annual incidence of 1–2 per 1000 individuals [1,2]. It is associated with serious short- and long-term complications including recurrence, post-thrombotic syndrome, chronic thromboembolic pulmonary hypertension, and death [1,2,3,4,5]. Despite improved awareness and advances in thromboprophylaxis, time trend studies have shown that the incidence of VTE has slightly increased over the past decades [6,7,8]. It is likely that the incidence of VTE will continue to rise, because major risk factors for VTE, such as advancing age, obesity, and the incidence of cancer, are increasing in the population [9,10,11]. To reduce the health burden associated with VTE, it is crucial to identify novel biomarkers and unravel molecular pathways involved in the pathogenesis of VTE in order to improve risk stratification and pursue targeted prevention and treatment.

MicroRNAs (miRNAs) are small endogenous non-coding RNAs consisting of approximately 22 nucleotides that downregulate gene expression at the post-transcriptional level through complementary base-pairing with the target messenger RNA (mRNA) [12]. They regulate gene expression either by translation inhibition or mRNA degradation [12]. Since the discovery of miRNAs as a class of regulatory molecules, their critical role in many biological and pathological processes has become apparent [13,14,15]. A key feature of the mechanism of action of miRNAs is that a single miRNA can regulate the expression of multiple genes, and, conversely, individual genes can be regulated by different miRNAs [12]. In addition to their common intracellular localization, a substantial number of miRNAs can be detected extracellularly in various body fluids including plasma and serum [16]. MiRNAs are transported in the extracellular environment by different carriers such as microvesicles, exosomes, and lipoproteins [17,18,19]. In human plasma, the binding to carriers makes miRNAs remarkably stable, as they are protected from endogenous RNase activity [20,21,22]. Given the stability and presence in various body fluids, miRNAs have emerged as a promising class of biomarkers for many diseases including cancer and cardiovascular diseases [14].

The expression of several hemostatic factors has been reported to be regulated by miRNAs [23,24] including key procoagulant factors and inhibitors of the coagulation pathway as well as plasminogen activator inhibitor-1, the main modulator of fibrinolysis [25,26,27,28,29,30,31,32,33,34,35,36,37] (Table 1 and Figure 1). In a recent study, Nourse et al. (2018) [38] revealed that individual genes encoding key hemostatic factors can be targeted by multiple miRNAs, suggesting that miRNAs act cooperatively in the regulation of the hemostatic system. MiRNAs may also play a relevant role in platelet biology with the potential to modulate platelet activation and aggregation as addressed by a comprehensive review on this topic [39]. Moreover, growing evidence from epidemiological studies and animal models suggests that the expression level of miRNAs is dysregulated in venous thrombosis. Taken together, miRNAs may be relevant not only as biomarkers for VTE but could also be involved in the pathogenesis of VTE and, therefore, be potential therapeutic targets. The aim of this narrative review is to summarize the current knowledge on the role of miRNAs as biomarkers for VTE and provide general insight into research exploring the modulation of miRNA activity in animal models of venous thrombosis. Methodological challenges and future perspectives in research on miRNAs and VTE will also be discussed.

The PubMed database was searched for publications on miRNAs and VTE by using combinations of the terms “microRNA”, “miRNA”, “miR”, “venous thrombosis”, “venous thromboembolism”, “deep vein thrombosis”, and “pulmonary embolism”. Relevant publications were also identified by cross-referencing from the reference lists of the retrieved papers.

## 2. Epidemiological Studies: miRNAs and Venous Thromboembolism

VTE risk prediction remains a challenge for clinicians, and despite previous efforts to identify predictive plasma biomarkers for VTE, only D-dimer is in clinical use for the diagnostic work-up of suspected acute VTE [40] and prediction of VTE recurrence [41]. D-dimer is a split product from the cross-linked fibrin clot, but it has low specificity of VTE since many other conditions such as cancer, inflammation, and pregnancy are associated with elevated D-dimer levels [42]. D-dimer has also the important drawback of being influenced by anticoagulant treatment. Therefore, there is a need to discover novel biomarkers for VTE capable of guiding clinical decision making. The number of studies exploring the contribution of miRNAs as VTE biomarkers has substantially increased during the past few years. However, most of the reports were case-control studies evaluating the potential use of miRNAs as biomarkers for the diagnosis of VTE as summarized in Table 2.

In 2011, Xiao et al. [43] assessed the expression profile of miRNAs in patients presenting with acute PE and found that plasma levels of miR-134 were higher in patients with acute PE (*n* = 32) compared to healthy controls (*n* = 32) or patients with cardiopulmonary diseases but without acute PE (*n* = 22). Four years later (2015), Qin et al. [44] measured serum miRNA expression levels after orthopedic surgery of the knee or hip in 38 subjects, of whom 18 had acute DVT and 20 had no evidence of DVT. They reported higher levels of miR-582, miR-195, and miR-532 in subjects with DVT compared to those without DVT [44]. In 2016, Wang et al. [45] investigated the miRNA expression levels in plasma of 238 patients with suspected DVT and found that levels of miR-424-5p were higher, whereas levels of miR-136-5p were lower in DVT patients compared to those without DVT. In the same year, Kessler et al. (2016) [46] reported that serum levels of miR-1233 were higher in patients presenting with acute PE (*n* = 30) compared to patients with acute non ST-segment elevation myocardial infarction (*n* = 30) or healthy controls (*n* = 12). In two other studies by Zhou et al. (2016) [47] and Liu et al. (2018) [48], plasma expression levels of miR-28-3p and miR-221 were shown to be upregulated in PE patients compared to healthy controls. In studies in which the expression of a specific miRNA was examined, levels of miR-26a [49] were found to be downregulated, whereas levels of miR-27a, miR-27b, miR-320a, and miR-320b were upregulated [50,51] in PE and DVT patients as described in Table 2.

Of note, some studies used epidemiological and experimental approaches in their analysis. For instance, Sahu et al. (2017) [27] searched for differentially expressed miRNAs in a rat model of DVT using inferior vena cava (IVC) ligation (IVC stasis model) and control animals and found that miR-145 was significantly downregulated in experimental DVT. Then, they tested the expression level of miR-145 in 20 male patients with VTE and 20 controls, and in line with their animal study findings, plasma miR-145 levels were lower in VTE patients compared to controls. Sun et al. (2020) [52] demonstrated that the expression level of miR-103a-3p was downregulated not only in patients with acute DVT (*n* = 81) versus healthy controls (*n* = 20) but also in a mouse model of DVT (IVC stenosis model). Zhang et al. (2020) [53] demonstrated that the expression of miR-338-5p was substantially downregulated in peripheral blood mononuclear cells of DVT patients (*n* = 36) in comparison to healthy controls (*n* = 36). Consistent with the findings in DVT patients, the expression of miR-338-5p was significantly lower in a mouse model of DVT (IVC stenosis model) versus control mice [53].

In contrast to studies that investigated the role of miRNAs as diagnostic biomarkers during the acute phase of a VTE, only a few studies addressed the role of miRNAs as predictive biomarkers for a first and recurrent VTE event. In 2015, Starikova et al. [54] used a case-control study derived from a population-based cohort (the Tromsø study) to evaluate the miRNA expression profile in the plasma of 20 patients with a first unprovoked VTE and 20 age- and sex-matched healthy controls. Patients were included in the study 1–5 years after the thrombotic event and at least three months after stopping anticoagulant and antiplatelet treatment. The study revealed that 5 miRNAs were upregulated, and 4 miRNAs were downregulated in VTE patients versus controls (Table 2). Wang et al. (2019) [55] were the first to examine whether circulating miRNAs were associated with recurrent VTE. The authors used a nested case-control study derived from the Malmö Thrombophilia Study, where the expression of miRNAs was quantified in plasma of 78 patients with unprovoked VTE two weeks after discontinuation of anticoagulation. Several miRNAs were differentially expressed in VTE patients with a recurrent event (*n* = 39) as compared to those without recurrence (*n* = 39) (Table 2). Finally, whether miRNAs could be used to improve risk prediction of VTE in cancer patients would be a clinically relevant question to address. Even though several risk stratification tools have been proposed to identify a subset of cancer patients at high risk of developing VTE, their clinical usefulness remains under debate [56]. It is of interest that in a recent study, Oto et al. (2020) [57] identified a profile consisting of 7 miRNAs (miR-486-5p, miR-106b-5p, let-7i-5p, let-7g-5p, miR-144-3p, miR-19a-3p, and miR-103a-3p) associated with the occurrence of a future VTE event during follow-up in a cohort of 32 patients with pancreatic ductal adenocarcinoma and distal extrahepatic cholangiocarcinoma.

### Methodological Challenges and Future Perspectives in Epidemiological Research on miRNAs and Venous Thromboembolism

It is noteworthy that among all miRNAs characterized in the fourteen studies described in Table 2, only six miRNAs were identified in more than one study. In the context of VTE, miR-532 [44,55], miR-320a [51,54], miR-320b [51,54], and miR-424-5p [45,54] were found to be upregulated, miR-103a-3p [52,54,55] was downregulated, and miR-27b yielded conflicting results, as it was up- and downregulated [50,55]. Several factors may account for the poor reproducibility among studies such as differences in clinical characteristics of participants, time from the occurrence of the thrombotic event to blood sampling, conditions of sample collection, sample processing (e.g., centrifugation speed and time) and storage, protocol used for RNA isolation, and methods for miRNA quantification and data normalization [20,58]. Another factor that may substantially contribute to the inconsistent findings is the limited statistical power of the individual studies due to the small sample sizes.

Recently, a meta-analysis assessed the predictive value of circulating miRNAs for the diagnosis of VTE [59]. Several miRNAs described in Table 2 (miR-1233, miR-134, miR-145, miR-582, miR-532, and miR-195) showed potential diagnostic value in this meta-analysis with an area under the receiver operating characteristic curve value > 0.8. However, the small sample size of the studies and the lack of external validation of the findings in larger cohorts represent major methodological concerns, and much effort is still needed to clarify the potential of miRNAs as diagnostic biomarkers for VTE.

Despite the progress of miRNA research in VTE over the past few years, there is still a lack of knowledge on the distinctive miRNA patterns specific for VTE in the general population. All the abovementioned studies (Table 2) were conducted using a case-control design, in which blood sampling took place after the thrombotic event. Therefore, the expression profile of these miRNAs could be merely a consequence rather than a mediator of the thrombotic disease. Clarification of this issue is worth pursuing, not only for the potential clinical use of miRNAs as predictive biomarkers for VTE but also as targets for novel therapeutics approaches. Next, we provide some general insight into current research exploring the impact of modulating miRNA activity on experimental DVT in animal models.

**Table 2 ijms-21-02602-t002:** Epidemiological studies on microRNAs (miRNAs) and venous thromboembolism (VTE).

First Author (Year)	Country	Study Population	Main Findings miRNA Expression Levels in Patients with VTE Versus Subjects without VTE	Potential Role of miRNA as Biomarker
Xiao et al. (2011) [43]	China	32 acute PE patients, mean age 54.8 ± 16.2 years, 47% men 22 non-acute PE patients, mean age 62.3 ± 23.3 years, 45% men ^1^ 32 healthy controls, NA: age and sex	↑ miR-134	Diagnostic biomarker for PE
Qin et al. (2015) [44]	China	18 acute postoperative DVT patients, mean age 69.4 ± 8.1 years, 28% men 20 postoperative control subjects, mean age 67.6 ± 7.2 years, 20% men	↑ miR-582, ↑ miR-195, **↑ miR-532**	Diagnostic biomarker for DVT
Starikova et al. (2015) [54]	Norway	20 patients with a history of first unprovoked VTE (1–5 years prior to inclusion in the study), mean age 56.4 ± 14.8 years, 50% men 20 healthy controls, mean age 56.3 ± 14.4 years, 50% men	↑ miR-10b-5p, **↑ miR-320a**, **↑ miR-320b**, **↑ miR−424-5p**, ↑ miR−423-5p **↓ miR-103a-3p**, ↓ miR−191-5p, ↓ miR−301a-3p, ↓ miR-199b-3p	Predictive biomarker for unprovoked VTE
Wang et al. (2016) [45]	Sweden	53 patients with DVT, mean age 59.8 ± 19.1 years, 40% men 185 patients without DVT, mean age 58.1 ± 16.8 years, 38% men	**↑ miR-424-5p**, ↓ miR-136–5p	Diagnostic biomarker for DVT
Kessler et al. (2016) [46]	Germany	30 acute PE patients, mean age 62.0 ± 14.0 years, 57% men 30 acute non-ST-segment elevation myocardial infarction patients, mean age 64.0 ± 13.0 years, 57% men 12 healthy controls, mean age 31.0 ± 6.0 years, 50% men	↑ miR-1233	Diagnostic biomarker for PE
Zhou et al. (2016) [47]	China	37 PE patients, mean age 42.0 ± 11.0 years, 57% men 37 healthy controls, mean age 41.0 ± 8.0 years, 54% men	↑ miR-28-3p	Diagnostic biomarker for PE
Sahu et al. (2017) [27]	India	20 VTE patients, median age 31.5 years, 100% men 20 controls, NA: age and sex	↓ miR-145	Diagnostic biomarker for VTE
Li et al. (2017) [49]	China	45 DVT patients with bone trauma, mean age 53 ± 8.6 years, 60% men 40 healthy controls, NA: age and sex	↓ miR-26a	Diagnostic biomarker for DVT
Wang et al. (2018) [50]	China	78 acute PE patients, mean age 61.0 ± 11.9 years, NA: sex 70 controls, mean age 62.0 ± 10.2 years, NA: sex	↑ miR-27a, ↑ miR-27b	Diagnostic biomarker for PE
Jiang et al. (2018) [51]	China	30 DVT patients, mean age 52.6 ± 15.4 years, 53% men 30 healthy controls, mean age 51.6 ± 12.7 years, 67% men	**↑ miR-320a**, **↑ miR-320b**	Diagnostic biomarker for DVT
Liu et al. (2018) [48]	China	60 acute PE patients, mean age 55.8 ± 7.5 years, 58% men 50 healthy controls, mean age 55.2 ± 7.0 years, 56% men	↑ miR-221	Diagnostic biomarker for PE
Sun et al. (2020) [52]	China	81 acute DVT patients, mean age 45.5 ± 9.1 years, 40% men 20 healthy controls, mean age 44.5 ± 6.8 years, 40% men	**↓ miR-103a-3p**	Diagnostic biomarker for DVT
Wang et al. (2019) [55]	Sweden	39 VTE patients with recurrent VTE (cases), median age 65.3 (IQR 11.7), 59% men 39 VTE patients without recurrent VTE (controls), median age 65.1 (IQR 11.9), 59% men	VTE patients with recurrence versus VTE patients without recurrence: ↑ miR-15b-5p, ↑ miR-222-3p, ↑ miR-26b-5p, **↑ miR-532-5p**, ↑ miR-21-5p, ↑ miR-30c-5p, ↑ miR-146b-5p, ↑ miR-22-3p ↓ miR-106a-5p, ↓ miR-197-3p, ↓ miR-652-3p, ↓ miR-361-5p, ↓ miR-27b-3p, **↓ miR-103a-3p**	Predictive biomarker for VTE recurrence
Zhang et al. (2020) [53]	China	36 DVT patients with symptom duration ≤ 21 days, mean age 57.3 ± 9.9 years, 47% men 36 healthy controls, mean age 54.1 ± 8.7 years, 44% men	↓ miR-338-5p	Diagnostic biomarker for DVT

DVT, deep vein thrombosis; NA, not available; IQR, interquartile range; PE, pulmonary embolism. ↑ miRNAs upregulated in VTE, DVT or PE. ↓ miRNAs downregulated in VTE, DVT or PE. ^1^ Non-acute PE patients: pneumonia (*n* = 7), unstable angina pectoris (*n* = 7), acute myocardial infarctions (*n* = 3), lung cancer (*n* = 2), pleurisy (*n* = 1), bronchiectasis (*n* = 1), asthma (*n* = 1) [43]. MiRNAs presented in bold have been shown to be upregulated (miR-532, miR-320a, miR-320b, and miR-424-5p) or downregulated (miR-103a-3p) in more than one study.

## 3. Modulation of miRNA Activity in Animal Models of Venous Thrombosis

Given the crucial role of miRNAs as regulators of many cellular and developmental processes and their dysregulation in several human diseases [13,14,15], miRNAs have emerged as attractive tools and targets for novel therapeutic approaches [60,61,62]. The use of miRNAs as a therapeutic modality essentially aims to restore or inhibit the function of a miRNA. Restoring the function of a miRNA that is downregulated in disease can be achieved by the use of miRNA mimics, which are synthetic, double-stranded, small RNA molecules that match the corresponding miRNA sequence [60,61,62]. Another approach to increase the level of a miRNA is to use viral vectors that over-express the miRNA of interest [60,61,62]. Contrary, the function of a miRNA that is upregulated in disease can be inhibited by the use of chemically modified, single-stranded antisense oligonucleotides, known as antimiRs [60,61,62].

MiRNAs display interesting features from a therapeutic viewpoint. For instance, the mature miRNA sequences are short and often completely conserved across multiple vertebrate species which make miRNAs relatively easy to target [61]. The use of miRNAs as a therapeutic modality has been investigated in many diseases in preclinical studies using animal models such as cancer, atherosclerosis, myocardial infarction, heart failure, infection by hepatitis C virus (HCV), kidney fibrosis, and diabetes [15,61,62]. Notably, there have been considerable advances in miRNA-based therapeutics with clinical trials being conducted (phases I/II), particularly in cancer and HCV infection [15,62].

In the context of VTE, a growing number of animal model studies have been published over the past few years exploring how the modulation of the activity of miRNAs may influence thrombogenesis and thrombus resolution as summarized in Table 3. We selected only those studies that used large vein models of venous thrombosis including IVC ligation or stasis models, in which the thrombi are formed in the absence of blood flow, and IVC stenosis model in which the thrombi are formed in the presence of blood flow [63]. Both models have been proven to be useful for studying venous thrombosis in vivo, mimicking the clinical scenario of occlusive (IVC stasis model) or nonocclusive (IVC stenosis model) venous thrombus [63]. Studies generally use bioinformatics analyses or previous findings on expression profile of miRNAs in DVT patients to select the miRNAs to be investigated in vivo (Table 3). Over-expression of miR-150, miR-126, let-7e-5p, and miR-21 by viral constructs or inhibition of miR-483-3p resulted in increased homing of endothelial progenitor cells and venous thrombus resolution in experimental DVT [64,65,66,67,68,69]. The target genes identified to be downregulated by the aforementioned miRNAs and potentially involved in the mechanism of thrombus resolution are described in Table 3. These genes have been implicated in the regulation of cell differentiation, migration, apoptosis, and angiogenesis [64,65,66,67,68,69].

Modulation of miRNA activity also seems to influence thrombogenesis (Table 3). Sahu et al. (2017) [27] demonstrated that replacement of miR-145 levels in thrombotic animals via systemic delivery of miR-145 mimics resulted in decreased expression of tissue factor and venous thrombus formation. In this study, bioinformatics tools and in vitro assays identified tissue factor as a target gene for miR-145. Interestingly, intravenous administration of miR-145 mimics was not associated with any detectable subacute toxicity or abnormal bleeding time in vivo [27]. Three years later, Sun et al. (2020) [52] demonstrated that over-expression of miR-103a-3p by viral construct resulted in decreased venous thrombus formation in experimental DVT. Such finding reinforces the potential role of miR-103a-3p in the pathogenesis of VTE, since this miRNA has also been found to be downregulated in VTE patients in three different studies [52,54,55] as previously described (Table 2). Zhang et al. (2020) [53] reported that intravenous administration of miR-338-5p mimics resulted in decreased interleukin-6 expression and venous thrombus formation. In vitro assays confirmed that miR-338-5p regulated interleukin-6 expression. Overall, results from Zhang et al. [53] suggest that decreased miR-338-5p may promote DVT formation by increasing interleukin-6 expression.

Notably, miRNAs may be relevant not only as biomarkers for VTE in cancer but may also play a role in the pathogenesis of cancer-related VTE. For instance, galectin-3 is a β-galactoside binding protein regulated by miR-322 that has been associated with the development of human cancers [70,71,72]. Galectin-3 has also been reported to be upregulated in a murine stasis model of venous thrombosis by DeRoo et al. (2015) [73] and may contribute to thrombosis via interleukin-6–dependent mechanisms [73]. Of note, in the same study, knocking out galectin-3 significantly reduced thrombus weight [73]. Altogether, these findings suggest that miRNA-related pathways have the potential to emerge as therapeutic targets for VTE prevention in cancer, and future research is necessary to unravel the complex inter-relation between miRNAs, cancer, and VTE.

## 4. Conclusions

Despite the progress of miRNA research in VTE over the past few years, the role of miRNAs as biomarkers for VTE has yet to be thoroughly established. As summarized in the present review, there is poor reproducibility among studies, with only a few miRNAs being identified in more than one study. The limited statistical power of the individual studies due to the small sample sizes and the lack of external validation of the study findings in larger cohorts still represent major methodological challenges in epidemiological research on miRNAs and VTE. Future studies addressing these limitations are warranted in order to elucidate the clinical usefulness of miRNAs as biomarkers for the diagnostic work-up of acute VTE and prediction of a first and recurrent VTE.

The potential role of miRNAs in the pathogenesis of VTE also deserves a deeper understanding. In animal model studies, over-expression or inhibition of some miRNAs have been suggested to influence venous thrombus formation and resolution. However, there is an important gap in knowledge regarding the potential role of miRNAs as therapeutic targets in VTE. Future research should also focus on developing rigorous preclinical studies in animal models to clarify the therapeutic properties and toxicity effects of miRNAs as a potential therapeutic modality of VTE.

## Figures and Tables

**Figure 1 ijms-21-02602-f001:**
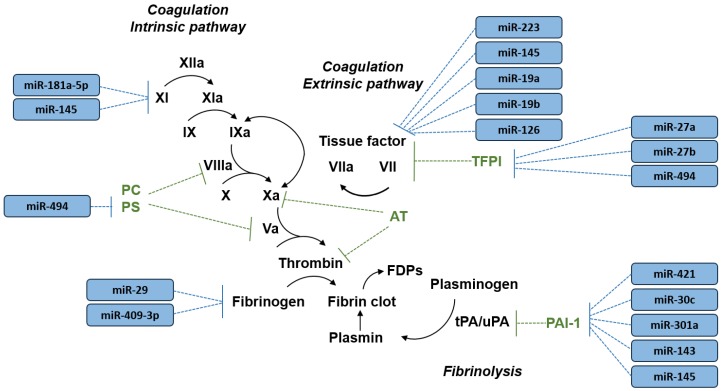
A simplified overview of blood coagulation and fibrinolysis pathways and the microRNAs (miRNAs) reported to regulate key hemostatic factors (see Table 1 for literature references). Black arrow indicates activation of hemostatic factors. AT, antithrombin; FDPs, fibrin degradation products; PC, protein C; PS, protein S; PAI-1, plasminogen activator inhibitor-1; TFPI, tissue factor pathway inhibitor; tPA, tissue plasminogen activator; uPA, urokinase plasminogen activator.

**Table 1 ijms-21-02602-t001:** MicroRNAs (miRNAs) reported to regulate hemostatic factors.

Target Gene	Protein	miRNA ^1^
*FGA, FGB, FGG*	Fibrinogen	miR-29 [25] miR-409-3p [25]
*F3*	Tissue factor	miR-223 [26] miR-145 [27] miR-19a [28] miR-19b [29] miR-126 [28,30]
*F11*	Factor XI	miR-181a-5p [31,32] miR-145 [32]
*PROS1*	Protein S	miR-494 [33]
*TFPI*	Tissue factor pathway inhibitor	miR-27a [34] miR-27b [34] miR-494 [34]
*SERPINE1*	Plasminogen activator inhibitor-1	miR-421 [35] miR-30c [35,36] miR-301a [36] miR-143 [37] miR-145 [37]

^1^ In vitro validation of interaction between the miRNA and target.

**Table 3 ijms-21-02602-t003:** MicroRNAs (miRNAs) and venous thrombosis in animal model studies.

First Author (year)	Country	Model Experimental DVT	miRNA Studied	Main Findings Impact of Modulation of miRNA Activity on Experimental DVT	Potential Target Genes Associated with the Mechanism of Experimental DVT ^1^
Wang et al. (2014) [64]	China	Rat model of DVT by IVC ligation (IVC stasis model)	miR-150	Intravenous injection of viral vector expressing miR-150 resulted in enhanced EPC homing and venous thrombus recanalization and resolution	*c-Myb* (c-Myb proto-oncogene)
Meng et al. (2015) [65]	China	Rat model of DVT by IVC ligation (IVC stasis model)	miR-126	Intravenous injection of viral vector expressing miR-126 resulted in enhanced EPC homing and venous thrombus recanalization and resolution	*PIK3R2* (phosphoinositide-3-kinase regulatory subunit 2)
Kong et al. (2016) [66]	China	Rat model of DVT by IVC ligation (IVC stasis model)	let-7e-5p	Intravenous injection of viral vector expressing let-7e-5p resulted in enhanced EPC homing and venous thrombus revascularization	*FASLG* (Fas ligand)
Kong et al. (2016) [67]	China	Rat model of DVT by IVC ligation (IVC stasis model)	miR-483-3p	Intravenous injection of viral vector expressing miR-483-3p inhibitor resulted in enhanced EPC homing and venous thrombus recanalization and resolution	*SRF* (serum response factor)
Sahu et al. (2017) [27]	India	Rat model of DVT by IVC ligation (IVC stasis model)	miR-145	Intravenous injection of miR-145 mimics resulted in decreased tissue factor mRNA levels and activity, and reduced venous thrombus formation	*F3* (coagulation factor III, tissue factor)
Wang et al. (2019) [68]	China	Rat model of DVT by IVC ligation (IVC stasis model)	miR-150	Intravenous injection of EPCs transfected with miR-150 mimics resulted in enhanced venous thrombus resolution	*SRCIN1* (SRC kinase signaling inhibitor 1)
Du et al. (2019) [69]	China	Rat model of DVT by IVC ligation (IVC stasis model)	miR-21	Injection within the thrombus of viral vector expressing miR-21 resulted in enhanced venous thrombus resolution	*FASLG* (Fas ligand)
Sun et al. (2020) [52]	China	Mouse model of DVT by IVC stenosis (IVC stenosis model)	miR-103a-3p	Intravenous injection of viral vector expressing miR-103a-3p resulted in decreased inflammatory cell infiltration and venous thrombus formation	*CXCL12* (C-X-C motif chemokine ligand 12)
Zhang et al. (2020) [53]	China	Mouse model of DVT by IVC stenosis (IVC stenosis model)	miR-338-5p	Intravenous injection of miR-338-5p mimics resulted in decreased interleukin-6 expression and venous thrombus formation	*IL6* (interleukin 6)

DVT, deep vein thrombosis; EPC, endothelial progenitor cells; IVC, inferior vena cava. ^1^ Experimental validation of potential target genes performed with luciferase reporter assay and/or Western blot analysis.

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
