# Peer review of "Role of microRNAs in Venous Thromboembolism"

_ijms, 2020, doi:10.3390/ijms21072602_

Round 1
Reviewer 1 Report
In their review, Morelli et al., described the role of miRNAs in venous thromboembolism context. In general the review is well written and structured. I have following concerns.
- The authors could describe the mechanisms of mi-RNA-based regulation of hemostatic factors, ie, HNF4, serpin.
- What is known about the correlation between miRNA, platelet, neutrophil activation markers?
- VTE in cancer patients is very problematic situation. A small paragraph could be added on cancer-VTE-miRNA. Lectins are regulated by miRNA PMID: 17889671, In venous thrombosis, the role of Galectin-3 and Galectin-3-binding proteins, PMID: 25428218, have been highlighted. In addition, recently it has been shown that Galectin-3- binds to platelets, induce platelet activation, PMID: 32040544. These studies can be highlighted in the text.
- What is known about the role of red blood cells in VTE?
Author Response
April 5, 2020
Dear Ms. Glinda He
Assistant Editor
International Journal of Molecular Sciences
Thank you for inviting us to resubmit the manuscript entitled “Role of microRNAs in venous thromboembolism”. Please note that we have submitted a revised manuscript using the "Track Changes" function in Microsoft Word to indicate changes.
We have replied to all comments and issues raised by the reviewers point by point, and revised the manuscript accordingly.
Please extend our thanks to the reviewers for their careful consideration of our manuscript. We think the reviewers’ comments and the subsequent revision have improved the quality of the manuscript, and we hope you will find it suitable for publication in International Journal of Molecular Sciences.
On behalf of the authors,
With best regards,
Vânia M. Morelli
K.G. Jebsen Thrombosis Research and Expertise Center (TREC)
Department of Clinical Medicine
UiT - The Arctic University of Norway, N-9037 Tromsø, Norway
Phone: +47 77625105; Fax: +47 77625105
E-mail: vania.m.morelli@uit.no
Comments from the reviewers
Reviewer 1
In their review, Morelli et al., described the role of miRNAs in venous thromboembolism context. In general the review is well written and structured. I have following concerns.
We would like to thank the reviewer for the comments.
1. The authors could describe the mechanisms of mi-RNA-based regulation of hemostatic factors, ie, HNF4, serpin.
Response: Our brief report on hemostatic factors and miRNAs serves to introduce the aims of this narrative review, which are (i) to summarize the current knowledge on the role of miRNAs as biomarkers for VTE, and (ii) provide general insight into research exploring the modulation of miRNA activity in animal models of venous thrombosis. As the focus of this review is on the role of miRNAs in VTE, mainly from an epidemiological viewpoint, we believe that a detailed description of the mechanisms of miRNA-based regulation of all hemostatic factors will be beyond the scope of this review, and move the focus away from the current aims. We would therefore prefer not to address the role of miRNAs in hemostasis in detail.
2. What is known about the correlation between miRNA, platelet, neutrophil activation markers?
Response: Thank you for this comment. Given that this review is essentially focused on providing an overview of the role of miRNAs in VTE, we have used a more general approach when referring to the association between miRNAs and platelet function, citing a comprehensive review on this topic by De Los Reyes-García et al. (Platelets 2019;30:803-808) in the introduction. In the revised manuscript, we have added the following sentence to the introduction section (page 2, lines 58-60):
“MiRNAs may also play a relevant role in platelet biology, with the potential to modulate platelet activation and aggregation, as addressed by a comprehensive review on this topic [39].”
Regarding the association between miRNAs and neutrophil activation markers, we believe that it is an interesting but complex topic that lies beyond the main aims of this review.
3. VTE in cancer patients is very problematic situation. A small paragraph could be added on cancer-VTE-miRNA. Lectins are regulated by miRNA PMID: 17889671, In venous thrombosis, the role of Galectin-3 and Galectin-3-binding proteins, PMID: 25428218, have been highlighted. In addition, recently it has been shown that Galectin-3- binds to platelets, induce platelet activation, PMID: 32040544. These studies can be highlighted in the text.
Response: We agree with the reviewer that VTE in cancer patients is a major clinical challenge. We would like to clarify that we already addressed the potential of miRNAs to improve risk prediction of VTE in cancer in section 2 of the narrative review (“Epidemiological studies - miRNAs and venous thromboembolism”), reading as follows (page 4, lines 135-142):
“Finally, whether miRNAs could be used to improve risk prediction of VTE in cancer patients would be a clinically relevant question to address. Even though several risk stratification tools have been proposed to identify a subset of cancer patients at high risk of developing VTE, their clinical usefulness remains under debate [56]. It is of interest that in a recent study, Oto et al. (2020) identified a profile consisting of 7 miRNAs (miR-486-5p, miR-106b-5p, let-7i-5p, let-7g-5p, miR-144-3p, miR-19a-3p and miR-103a-3p) associated with the occurrence of a future VTE event during follow-up in a cohort of 32 patients with pancreatic ductal adenocarcinoma and distal extrahepatic cholangiocarcinoma [57].
As suggested by the reviewer, we have now added a paragraph on the possible contribution of miRNAs in the pathogenesis of cancer-related VTE, highlighting some of the interesting studies pointed out by the reviewer. We included this paragraph in section 3 of the revised manuscript (“Modulation of miRNA activity in animal models of venous thrombosis”), reading as follows (page 11, lines 241-249):
“Notably, miRNAs may be relevant not only as biomarkers for VTE in cancer, but may also play a role in the pathogenesis of cancer-related VTE. For instance, galectin-3 is a β-galactoside binding protein regulated by miR-322 that has been associated with the development of human cancers [70- 72]. Galectin-3 has also been reported to be upregulated in a murine stasis model of venous thrombosis by DeRoo et al. (2015) [73], and may contribute to thrombosis via interleukin-6–dependent mechanisms [73]. Of note, in the same study, knocking out galectin-3 significantly reduced thrombus weight [73]. Altogether, these findings suggest that miRNA-related pathways have the potential to emerge as therapeutic targets for VTE prevention in cancer, and future research is necessary to unravel the complex inter-relation between miRNAs, cancer and VTE.”
4. What is known about the role of red blood cells in VTE?
Response: Red blood cells may contribute to VTE, as suggested by epidemiological studies, showing that both quantitative and qualitative abnormalities of red blood cells (e.g. altered hematocrit, sickle cell disease and hemolytic anemias) are associated with VTE (Byrnes & Wolberg. Blood 2017;130:1795-1799). There are also several proposed mechanisms to explain this association, such as promotion of increased blood viscosity and thrombin generation within the thrombus (Byrnes & Wolberg. Blood 2017;130:1795-1799). We acknowledge that the role of red blood cells in VTE is relevant from both mechanistic and clinical viewpoints. However, we think that the role of red blood cells in VTE is not within the proposed framework of this review, which is essentially focused on the role of miRNAs in VTE. We would prefer not to address the role of red blood cell in VTE in this review, as it would move the focus away from our main aims.
Reviewer 2 Report
The review of Morelli and colleagues gives a good overview of the role of miRNAs on the haemostatic system, in particular regarding venous thromboembolism.
In table 1 and figure 1 the roles of miRNAs on haemostasis are reported. Since this part is more general some publications as the one from Nourse et al (J Thromb Haemost 2018 ; 16 : 2233-45) may be useful.
Concerning the rest of the review I see no major article missing.
Author Response
April 5, 2020
Dear Ms. Glinda He
Assistant Editor
International Journal of Molecular Sciences
Thank you for inviting us to resubmit the manuscript entitled “Role of microRNAs in venous thromboembolism”. Please note that we have submitted a revised manuscript using the "Track Changes" function in Microsoft Word to indicate changes.
We have replied to all comments and issues raised by the reviewers point by point, and revised the manuscript accordingly.
Please extend our thanks to the reviewers for their careful consideration of our manuscript. We think the reviewers’ comments and the subsequent revision have improved the quality of the manuscript, and we hope you will find it suitable for publication in International Journal of Molecular Sciences.
On behalf of the authors,
With best regards,
Vânia M. Morelli
K.G. Jebsen Thrombosis Research and Expertise Center (TREC)
Department of Clinical Medicine
UiT - The Arctic University of Norway, N-9037 Tromsø, Norway
Phone: +47 77625105; Fax: +47 77625105
E-mail: vania.m.morelli@uit.no
Comments from the reviewers
Reviewer 2
The review of Morelli and colleagues gives a good overview of the role of miRNAs on the haemostatic system, in particular regarding venous thromboembolism.
We would like to thank the reviewer for the comments.
In table 1 and figure 1 the roles of miRNAs on haemostasis are reported. Since this part is more general some publications as the one from Nourse et al (J Thromb Haemost 2018 ; 16 : 2233-45) may be useful.
Concerning the rest of the review I see no major article missing.
Response: As suggested by the reviewer, we have now added this interesting and relevant study on the role of miRNAs in the hemostatic system by Nourse et al. (J Thromb Haemost 2018; 16: 2233–45) to the revised manuscript. As our introduction on hemostatic factors and miRNAs serves to provide a general overview of the topic and not a detailed description, we included a general statement regarding the main findings of the above-mentioned study, reading as follows (page 2, lines 56-58):
“In a recent study, Nourse et al. (2018) [38] revealed that individual genes encoding key hemostatic factors can be targeted by multiple miRNAs, suggesting that miRNAs act cooperatively in the regulation of the hemostatic system.”
Reviewer 3 Report
A very well written, concise and comprehensive overview of this important topic and the potential role of miRNA in the aetiology of DVT. The article is pithy and well supported by relevant referencing and key studies synopsised and tabulated for the readers clarify and information.
One point to note- and this is a personal suggestion, would be to reference miRNA and platelet function- an area which is advancing greatly and of major interest. This I believe would be additive to the this nice review.
Author Response
April 5, 2020
Dear Ms. Glinda He
Assistant Editor
International Journal of Molecular Sciences
Thank you for inviting us to resubmit the manuscript entitled “Role of microRNAs in venous thromboembolism”. Please note that we have submitted a revised manuscript using the "Track Changes" function in Microsoft Word to indicate changes.
We have replied to all comments and issues raised by the reviewers point by point, and revised the manuscript accordingly.
Please extend our thanks to the reviewers for their careful consideration of our manuscript. We think the reviewers’ comments and the subsequent revision have improved the quality of the manuscript, and we hope you will find it suitable for publication in International Journal of Molecular Sciences.
On behalf of the authors,
With best regards,
Vânia M. Morelli
K.G. Jebsen Thrombosis Research and Expertise Center (TREC)
Department of Clinical Medicine
UiT - The Arctic University of Norway, N-9037 Tromsø, Norway
Phone: +47 77625105; Fax: +47 77625105
E-mail: vania.m.morelli@uit.no
Comments from the reviewers
Reviewer 3
A very well written, concise and comprehensive overview of this important topic and the potential role of miRNA in the aetiology of DVT. The article is pithy and well supported by relevant referencing and key studies synopsised and tabulated for the readers clarify and information.
We would like to thank the reviewer for the comments.
One point to note- and this is a personal suggestion, would be to reference miRNA and platelet function- an area which is advancing greatly and of major interest. This I believe would be additive to the this nice review.
Response: We agree with the reviewer that regulation of platelet function by mRNAs is an area of great interest and clinical relevance for bleeding and thrombotic diseases. Given that this manuscript is essentially focused on providing an overview of the role of miRNAs in VTE, we have used a more general approach in our reference to miRNA and platelet function, citing a recent and comprehensive review on this topic by De Los Reyes-García et al. (Platelets 2019;30:803-808). In the revised manuscript, we added the following sentence to the introduction section (page 2, lines 58-60):
“MiRNAs may also play a relevant role in platelet biology, with the potential to modulate platelet activation and aggregation, as addressed by a comprehensive review on this topic [39].”